# On the relation between COVID-19, mobility, and the stock market

**Robin Enrico van Ruitenbeek**[1☯]*, **Jesper Siem Slik**[1☯]*, **Sandjai Bhulai**[2]*

**1** Pon, Amsterdam, The Netherlands, **2** Vrije Universiteit Amsterdam, Amsterdam, The Netherlands

☯ These authors contributed equally to this work.
* r.e.van.ruitenbeek@vu.nl (RER); j.s.slik@vu.nl (JSS); s.bhulai@vu.nl (SB)

## Abstract

The Covid-19 pandemic has brought forth a major landscape shock in the mobility sector. Due to its recentness, researchers have just started studying and understanding the implications of this crisis on mobility. We contribute by combining mobility data from various sources to bring a novel angle to understanding mobility patterns during Covid-19. The goal is to expose relations between mobility and Covid-19 variables and understand them by using our data. This is crucial information for governments to understand and address the underlying root causes of the impact.

## 1 Introduction

One of the first visible impacts of the Covid-19 crisis was on transport, travel, and mobility. Mobility explained a substantial proportion of variance in transmissibility [1]. The travel restrictions adopted to limit the spread of the disease led to drastic reductions in travel and traffic. This had various implications. The disruption in the flow of goods had severe economic consequences. The measures on mobility, traffic, and transport also had a substantial impact on the socio-economic sector.

The crisis has affected all forms of transport, from bicycles, cars, public transport, maritime vessels, trains, and air flights [2]. The activity on global road transport was almost 50% below the 2019 average by the end of March. Similarly, commercial flights were almost 75% below by mid-April 2020 [3], while the global flight network density reduced by 51% [4]. Therefore, a key question is how changes in transport behavior affect each other due to Covid-19 and how they relate to the economic progression worldwide.

In the recent past, there have been a number of crises that have caused major changes in mobility patterns. For example, the Severe Acute Respiratory Syndrome (SARS) crisis of 2003 significantly affected air traffic in specific regions. The volume of traffic dropped by 35% [5]. Also, the non-essential trips with public transport dropped by 50% during the peak of the pandemic [6]. It took almost four months for the passenger numbers to return to pre-crisis levels.

The Avian Flu outbreaks of 2005 and 2013 and the Middle Eastern Respiratory Syndrome in 2015 also significantly impacted mobility. The demand for air travel in these cases returned back relatively quickly [5]. It is reasonable to assume that mobility patterns of the Covid-19 disease will be more in line with the patterns of SARS. The Covid-19 and SARS pandemics

**Data Availability Statement:** Data are held in a public repository, accessible through figshare. DOI's are available in the manuscript, the project containing all datasets is accessible through: https://figshare.com/projects/On_the_Relation_

between_Covid-19_Mobility_and_the_Stock_
Market/126397.

**Funding:** The authors received no specific funding
for this work.

**Competing interests:** The authors have declared
that no competing interests exist.

share the scale of the impact and the perceived risks of contagion, which are more significant
than other recent pandemics.

From a behavioral perspective, it is interesting to study the patterns in mobility and the
short- and long-term effects. This is crucial information for policymakers to understand and
address the underlying root causes of the impact. For example, after the terrorist attacks on 9/
11, there was a drop in air traffic demand that lasted five years after the attacks [7]. Studies
contribute this to the risks and inconvenience of flying after new security precautions were
introduced.

The Covid-19 crisis could bring forth different changes than other crises in the past. Busi-
ness travel could be replaced by more video conferencing since the technology has rapidly
matured in a short period of time [8]. Reduction of demand for particular modes of transport
could become permanent due to perceived risk [9]. A model shift could happen to modes of
transport that avoid contact with people to have less perceived exposure to the virus [10].
Thus, a model shift could happen from public transport to bicycles [11]. Governments can use
this information in change campaigns to change public behavior. This can influence which
transport behaviors are more permanent after the crisis.

In this paper, we combine mobility data from various sources to bring a novel angle to
understanding mobility patterns during Covid-19. We look at mobility data from bicycles,
maritime vessels, trains, car traffic, and air flights. First, we look at patterns in between these
modalities. Second, we relate the patterns to the Covid-19 cases and measures, as well as the
stock market. The goal is to expose relations between mobility and Covid-19 variables and
understand them by using our data.

The rest of this paper is structured as follows. In Section 2, we explain how we obtained the
data from the various data sources. We discuss our methodology for the analysis of the data in
Section 3. The results of the analysis are presented in Section 4. We conclude the paper in Sec-
tion 5 with a discussion on our research.

## 2 Data

We gather data from various sources to answer our research questions. In this section, we
describe each source and give an initial insight into its contents. The sources are related to the
usage of mobility, economic indicators, and Covid-19 statistics. All sources are on a global
scale, covering countries on all continents except Antarctica. The mobility types we consider
are vessels, flights, vehicles, trains, and bicycles. The economic indicators are extracted from
various stock markets and Covid-19 data from official numbers by the corresponding coun-
tries. The sources are available on a daily level. However, to account for intra-week seasonality,
we aggregate all sources to a weekly level to perform our analysis. Additionally, in the current
section only, most graphs show the percentage change since the first known value in the corre-
sponding time range, on a weekly level, smoothed over four weeks. We apply smoothing to
apply visual focus on the general trend and take into account the percentage change to create a
fair comparison amongst the different continents. All data is made publicly available through
[12–17].

### 2.1 Vessels

We use the real-time ais data from AISHub [18], which we collected over a time period rang-
ing from *2019-04-01* to *2020-12-01*. The raw data is collected with an interval of 2-3 seconds.
Using more than 600 ais stations, a total of 8.5 billion records of 4.8 million unique ship IDs
have been retrieved. We focus on commercial and cargo ships, including passenger ships,
tankers, cargo, and fishing. We exclude ship types such as military, medical, and towing.

To minimize the size of the data, we reduce the dataset to a single record per hour and use this dataset to estimate the vessel activity. We define the vessel activity for a port in time period $t$ as the number of vessels entering and exiting the port. To approximate if a vessel is inside a port, we assume a vessel $i$ to be in port $j$ if $\sqrt{\left(x_i - x_j\right)^2 + \left(y_i - y_j\right)^2} \leq \beta$ holds. For this, $(x_i, y_i)$ and $(x_j, y_j)$ are the longitude and latitude pairs of both the vessel location and the center of the port. We approximate the visits using the port center coordinates of [19] instead of port shape files to speed up the computation, since it has to be performed on all records of the extensive dataset. From this, we track each vessel over time and assign a port visit if the ship has been in the port for at least $\alpha = 3$ consecutive hours. We set $\alpha$ to 3 as bulk carriers and oil tankers move with the lowest average speed of 24 kilometers an hour [20], indicating that at least 2 hours is required to pass the port without entering it for $\beta = 12$ kilometers.

To finalize the vessel activity, we solely look at the data points where a vessel is within the range of a port. From this, we assign an entrance activity if the vessel moves into the range of a new port $j$ at time $t$ and is detected in the same port on (or after) $t + 3$ without visiting another port in between. Therefore, if a vessel is only detected once, it is assumed to be passing the port. Similarly, we define a port exit when the vessel meets the entrance criteria and is seen in another port after $t + 3$. We assign the exit time based on the last recorded time where vessel $i$ was seen in port $j$. With this, the defined vessel activity is relatively robust against sensor downtime, which frequently occurs due to vessels shutting down the system within a port.

Fig 1 visualizes the growth in vessel activity compared to the first week of January in 2020. We can observe large differences amongst the continents. Most notable is South America, which has generally been decreasing throughout the year. Most other continents show a dip around March and April. However, in November, they are close to their original value in January.

## 2.2 Flights

Similar to vessels, airplanes can be tracked in flight and on the ground through surveillance technology. The data we gather regarding flights originates from the system named ADS-B, as described by [21]. We collect daily statistics on multiple airports throughout the world, using two sources.

Our primary source is the AeroDataBox API, which is available through [22]. This API collects data from external public data sources, community-maintained and commercial

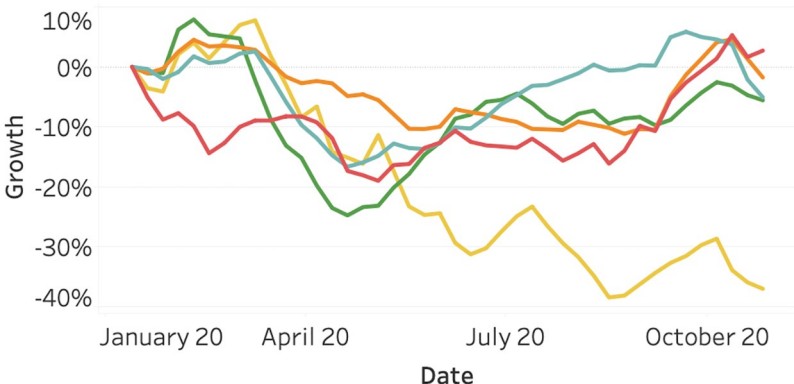

**Fig 1. Vessel activity as of January 2020, split per continent.** Port enters and exits are included in the data to account for intercontinental ships, exiting one continent and entering another. North America (orange), South America (yellow), Asia (red), Europe (light blue), and Oceania (green).

databases. Their collected data can be queried through an API made available on the RapidAPI platform.

Our secondary source is FlightRadar24, which can be accessed on their website through [23]. They describe themselves as a global flight tracking service, collecting real-time data on thousands of air crafts.

We do not track all flights in the world, however. We focus on departures from commercial, cargo, and private flights. By focusing on departures, we attribute each flight to the airport, and hereby country and continent from which it departs. Also, we attribute the date and time to the departure time. This prevents counting flights twice and, as the plane will generally depart from its arrival airport at a later time, has a minimal impact on misclassifying flights to the correct country and timestamp. We exclude canceled flights and count code-shared flights as a single flight (in case one flight has multiple operators).

Regarding the tracked locations, we selected 24 major airports over the world. We did so by analyzing our collected data. An estimate of the flight volume per airport is also available through [24]. We selected a maximum of 5 airports per continent, a maximum of 3 airports per country, and a minimum of 20 million passenger volume in 2018. S1 Text lists all ICAO codes of the airports we track.

Fig 2 visualizes the flights data. First, we observe that the starting date of the graph lies around March. The first data point in our database is the 24<sup>th</sup> of February. Second, we observe slight differences between the continents. Asia shows a relatively large number of flights since February, which can be explained by the fact that the Covid-19 outbreak started earlier. We possibly miss the part of the initial decline. All other continents show a large decline in flight activity after the outbreak in March.

## 2.3 Vehicles

We utilized connected vehicle data from 57 countries, distributed over the different continents, with the majority of data collected in America, Asia, and Europe. The data is aggregated on a daily level, where we introduce the traffic intensity on a daily level as a representation of the number of active vehicles during the day. In other words, the activity during time window $t$ is approximated by the average number of vehicles that operate during this time window. To reliably estimate the traffic intensity, we solely focus on cities where we selected 400 large cities distributed over the 57 countries.

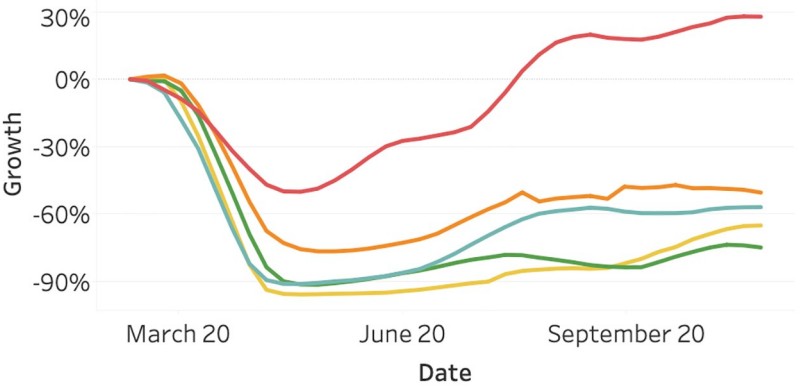

**Fig 2. Flights activity as of March 2020, split per continent.** North America (orange), South America (yellow), Asia (red), Europe (light blue), and Oceania (green).

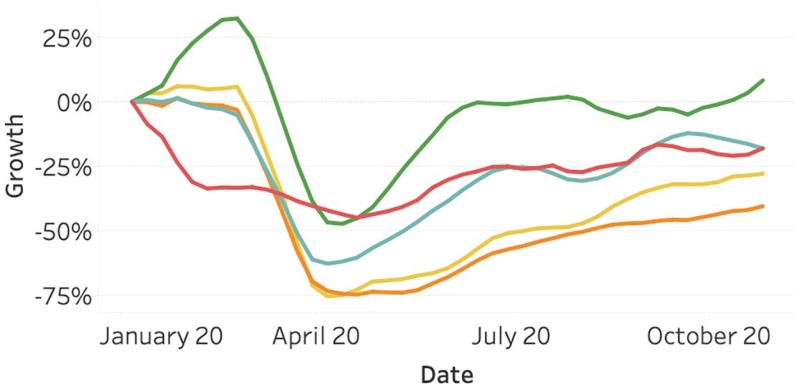

**Fig 3. Vehicle activity as of January 2020, split per continent.** North America (orange), South America (yellow), Asia (red), Europe (light blue), and Oceania (green).

Fig 3 visualizes the vehicle activity data. We observe a similar pattern to the flight activity. Most continents show a decline in vehicle activity in March, with the exception of Asia. After April, the traffic intensity of all continents is generally increasing.

## 2.4 Train and bicycle search activity

Besides ships, airplanes, and vehicles, two remaining and major transport types are trains and bicycles. However, gathering data on these sources is challenging on a global scale.

Typically, usage of bicycles is hardly registered, as they rarely contain sensors registering their usage. The electrification in the bike industry might change this, but it is currently not available. A potential source could be fitness tracking apps. However, they typically focus on exercising and performance but hardly on commuting. Besides, they do not publish their data for research purposes.

Train usage is challenging to retrieve as this is highly dependent on the respective operator. Different countries typically have different operators, which typically have a different method or granularity on which they publish data if they decide to publish these data. Given this research's global scope, we decided to find an alternative source to estimate train usage.

We estimate the usage of bicycles and trains by observing online search behavior. Google publishes this information on a global scale at [25]. Using this tool, we extract relative indices for online search behavior through Google on the topics bicycle and trains. We can do this over a time span of 2020, split by many countries in the world.

Fig 4 visualizes the Google search activity for the topic trains. Surprisingly, the impact of the Covid-19 outbreak on the search activity seems to be limited in South America, compared to the other continents. Additionally, in Europe, the activity grew to nearly the level of January 2020. However, it started decreasing again after August. This can be explained by the second peak of Covid-19.

Fig 5 visualizes the Google search activity for the topic bicycles. This seems to be the only mobility-related data source that has increased in activity after the first Covid-19 peak in 2020. For all continents, there is a positive growth relative to January. This growth is most notable in North America and Europe, which might partially be influenced by the seasonality. In summer, we expect more bicycle searches. However, both South America and Oceania show a rising search volume as of April 2020, despite the ending of summer. This growth might be explained by the decrease in public transport availability. Also, cycling is one of the few

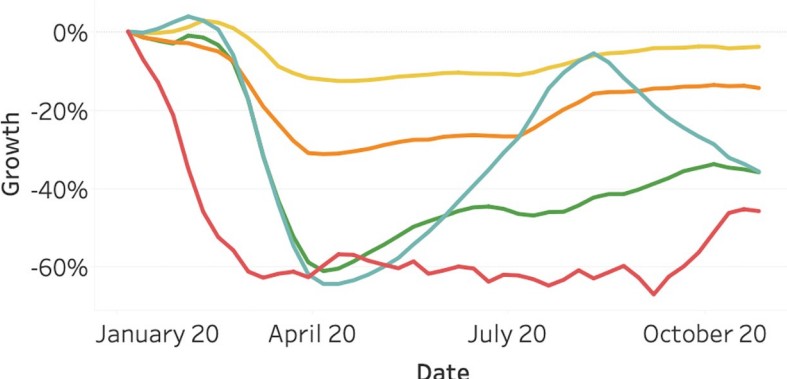

**Fig 4. Online searches for train through Google as of January 2020, split per continent.** North America (orange), South America (yellow), Asia (red), Europe (light blue), and Oceania.

outdoor activities that are possible under most Covid-19 related measures. People that usually practice team sports like football, basketball, or field hockey might have switched to cycling.

## 2.5 Stock markets

To obtain stock market information, we use the software implementation of [26]. For this, we tracked 40 major country indices, 74 raw materials, 566 stocks composed of the top-valued companies per index, and 148 currencies, of which 98 cryptocurrencies. The country indices are select from the Yahoo major world index list [27], where we added the largest index of The Netherlands, Austria, Sweden, and Spain to increase the coverage in Europe. All stock information is tracked over a time span from *2020-01-01* to *2020-11-12*. We transformed all market close prices on a daily level from its listed currency to usd, using the close exchange rate between the foreign currency and usd.

Fig 6 visualizes the average stock index growth per continent. We can observe a joint decline for all continents, where Oceania reacts slightly slower and recovers faster compared to the other continents. After the 20$^{\text{th}}$ of March, the stock indices on all continents are generally increasing.

Fig 7 presents the average stock index growth for various groups of raw materials. Apart from the energy sector, all material groups decline less extensively compared to the average

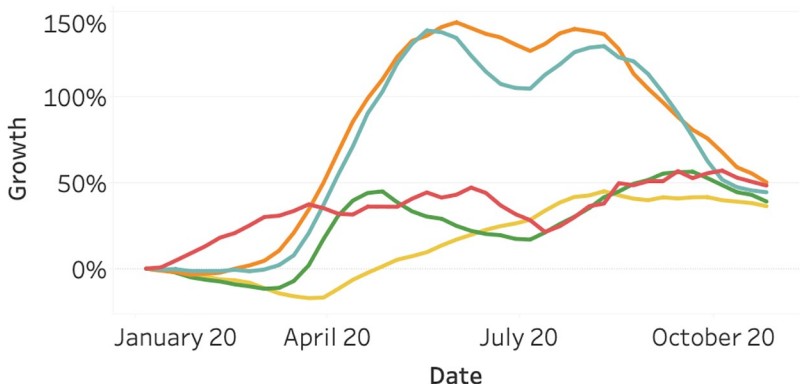

**Fig 5. Online searches for bicycle through Google as of January 2020, split per continent.** North America (orange), South America (yellow), Asia (red), Europe (light blue), and Oceania (green).

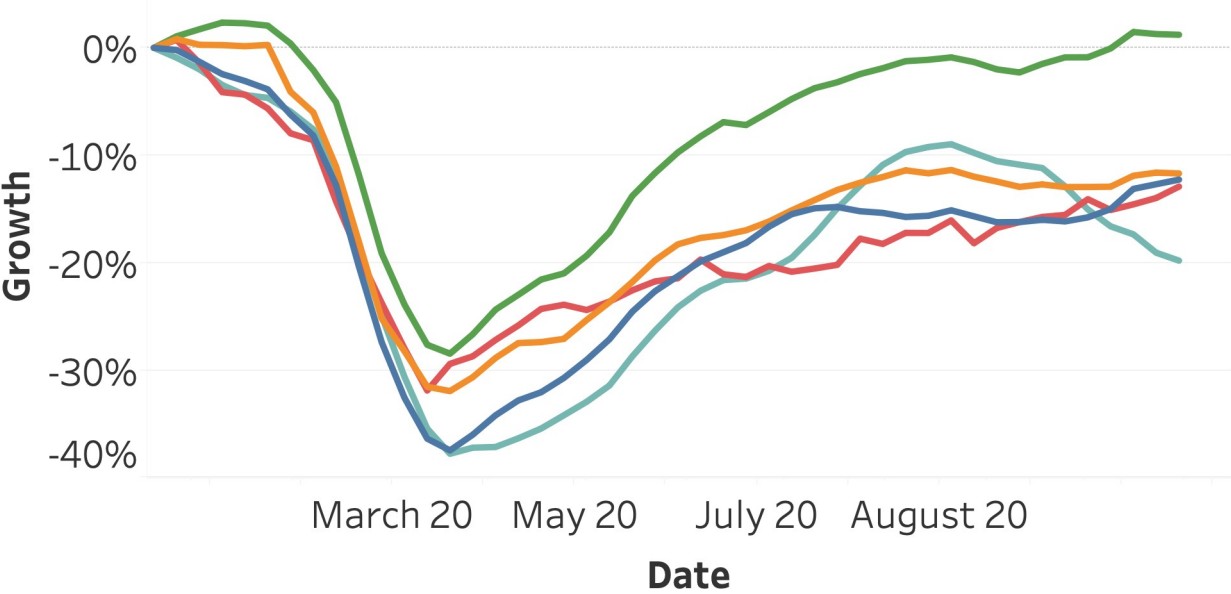

**Fig 6. Average stock indices growth as of January 2020, split per continent.** Africa (dark blue), America (orange), Asia (red), Europe (light blue), and Oceania (Green).

index per continent (Fig 6). The energy category suffers from a strong decline of more than 50 percent. Metals experience the lowest decline and the earliest recovery.

## 2.6 Covid-19 cases

One of the most central datasets of this research is the global Covid-19 dataset. Our primary source is the Covid-19 data repository of [28], sourced by the Center of Systems Science and

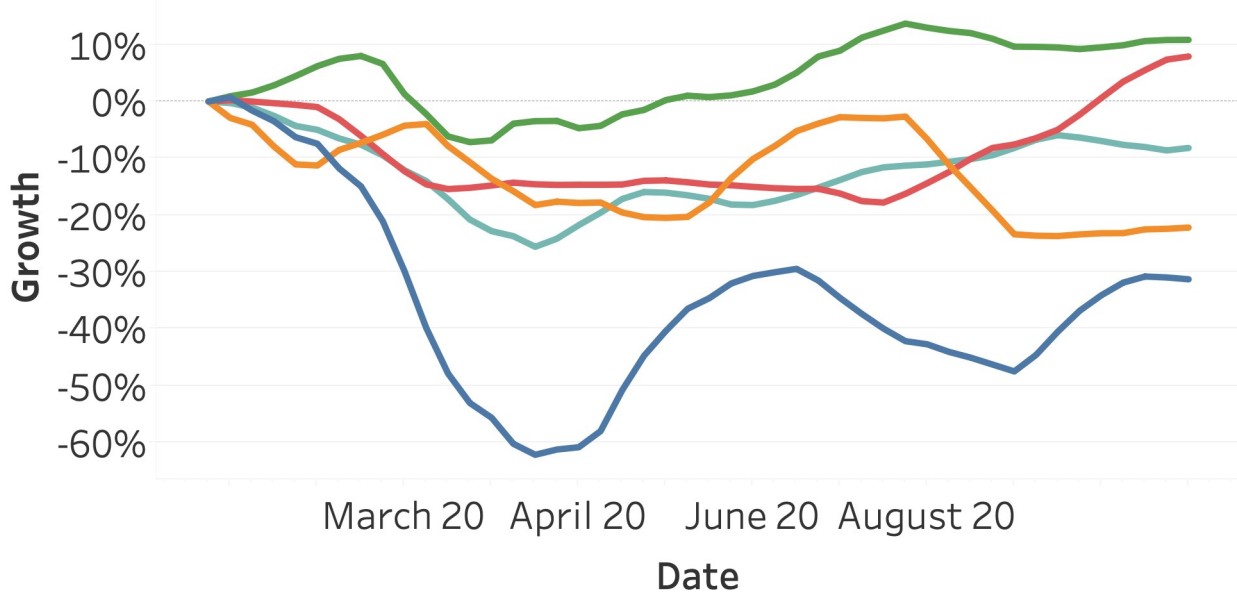

**Fig 7. Average raw material growth as of January 2020, split per category.** Energy (dark blue), Food & Fiber (orange), Grains (red), Livestock & Meets (light blue), and Metals (Green).

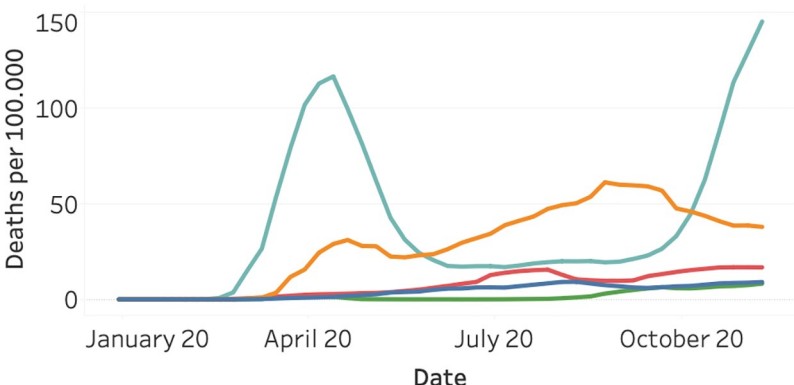

**Fig 8. Covid-19 deaths per 100,000 inhabitants as of January 2020, split per continent.** North America (orange), South America (yellow), Asia (red), Europe (light blue), and Oceania (Green).

Engineering (CSSE) at Johns Hopkins University. We extrapolated the absolute registered corona cases and related deaths per country on a daily level. We normalized the absolute deaths to deaths per 100,000 inhabitants to account for strong differences of inhabitants per country.

Fig 8 visualizes the normalized death per continent on a weekly level. Europe shows a strong first peak around the end of April and a second one at the start of November. Surprisingly, Europe shows a steep increase, followed by a strong decline in the number of deaths. Contrary to Europe, North America shows more consistent growth in the number of deaths and less intense waves. Surprisingly, South America, Asia, and Oceania observe far fewer deaths. This might partially be explained by differences in measuring and registering Covid-19 deaths.

### 2.7 Covid-19 measures

The measures to fight Covid-19 are strongly varying over time and between different countries or regions. To the best of our knowledge, there is no universal dataset available with all corona measures on a country level. Therefore, we limit the data collection of Covid-19 measures to The Netherlands in isolation. For this, we mainly extrapolated the measures from the different press conferences, using the website of the Dutch National Institute for Health (RIVM) [29].

From this, we obtained the following features: press conference date, the date the measures take effect, the opening or closure of the primary schools, the secondary schools, the universities, indoor sports, outdoor sports, and professions with close contact such as hairdressers. In addition, we construct features for the number of allowed visitors in restaurants, churches, home settings, and public spaces such as concert halls and theaters.

Fig 9 visualizes the opening (green) and mandatory closure (blue) for different segments.

### 3 Methodology

To answer our research questions, we split our methodology into three parts. First, we combine the various mobility-related data sources into one dataset. Second, we add corona and stock measures and investigate various relations in the resulting dataset. Third, we investigate whether we can quantify the impact of the Covid-19 measures taken in the Netherlands.

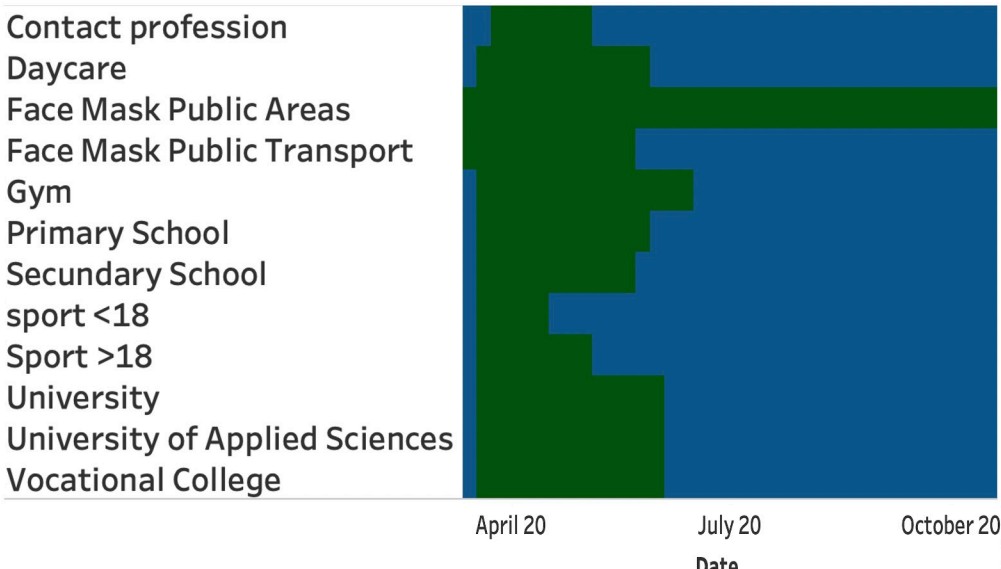

**Fig 9. Opening and mandatory closure for different business segments.** Opening (green), and mandatory closure (blue).

## 3.1 Combining mobilities

Combining the mobility-related data sources is a challenge, as their origin and data structures are not aligned. We tackle this by defining a base dataset to which all sources can be mapped. This dataset consists of the dimensions date and country. We apply filters corresponding to the most limited dataset, which in our case concerns flights. This dataset contains dates starting from the end of February, and is limited to a selected number of airports and hereby countries. However, we still cover major countries and the largest part of 2020. We aggregate the various sources to a daily and country level, average the corresponding measure, and merge the result to the base dataset. The resulting dataset will contain for each country and date the vessel, flight, vehicle, bike, and train measures. To adjust for weekly seasonality, we aggregate this set to a weekly level.

## 3.2 Relation between variables

One of the goals of this paper is to find relationships between all statistics gathered in this research. We do this by combining the data in an appropriate table, preprocessing its contents, and applying a correlation test and dynamic time warping to it.

First, we expand the mobility dataset we created in subsection 3.1 with corona cases, corona deaths, and stock indices. We filter the stock data only to contain country-related indices and average all indices within one country.

Second, we preprocess the dataset. We standardize the measures to having a mean of 0 and a standard deviation of 1. In addition, we compute lag variables. For each measure, we compute $measure\_lag\_i$ with $i \in \{1, 2, ..., 5\}$, in which the corresponding measure is lagged by $i$ weeks.

Finally, we investigate the relation between the resulting measures in terms of Pearson correlation and dynamic time warping. As described in [30], dynamic time warping is a robust distance metric that allows an elastic shifting of the time axis, to accommodate sequences that are similar. It compares points between two sequences in a many-to-one fashion, in contrast

to the one-to-one fashion of the Pearson correlation. We implemented this procedure through the Python implementation of [31].

### 3.3 Impact of Covid-19 measures

Quantifying the exact impact of Covid-19 measures is challenging as we solely have the Dutch measures available, and many other causal effects play a role. We aim to provide insights into the relation between corona deaths, mobility usage, and corona measures. The constructed dataset contains the maximum allowed visitors for specific sectors and presents which sectors are closed or open. Due to the limited sample size, a statistical test will not provide reliable and conclusive results. Therefore, we aggregate all data on a weekly level and visualize the major changes in corona measures within the chart. With this, we aim to present potential relationships between the three variables.

## 4 Results

In this section, we highlight our results in the same structure as described in the Methodology.

### 4.1 Combining mobilities

Fig 10 highlights the growth in the measured usage of the monitored mobility types, relative to the first of March 2020. The usage is measured in the manner described before and averaged on a global scale. We can observe the usage of all mobility types, except bicycles, has declined since March 2020. Flights show the largest decline, followed by traffic and train. Vessel activity has not decreased much—all in large contrast to the usage of bicycles, which shows a large increase in usage.

### 4.2 Relation between variables

Table 1 presents the strongest Pearson correlation coefficient, between the original variable on the rows and the lagged variable on the columns. All variables on the diagonal, as well as any non-significant ($p > 0.05$) variables, are omitted from the table. We can observe some strong correlations across all variables where most are significant. We see, for example, that both trains with lag 3 (columns) and traffic with lag 1 (columns) highly correlate with flights (rows).

More interestingly, we can observe that corona deaths seem to correlate more with the other variables, compared with corona cases. In addition, corona-related deaths seem to have a

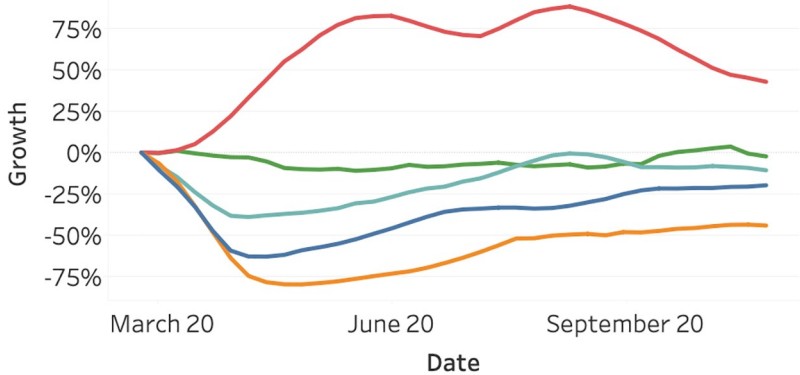

**Fig 10. Growth in global usage of mobility relative to March 2020, split per mobility type.** Bicycle (red), vessel (green), train (light blue), traffic (blue), and flights (orange).

**Table 1. Original time-series (lag = 0) on rows and lagged variable on columns.** The diagonal, as well as correlations that are not significant ($p > 0.05$) are omitted from the table. We present each correlation along with the best lag for the column variable (round brackets) and its significance level: a ($p < 0.05$), b ($p < 0.01$), c ($p < 0.001$) (superscript).

| | Traffic | Cases | Deaths | Flights | Bicycles | Train | Stock | Vessel |
|---|---|---|---|---|---|---|---|---|
| Traffic | | $0.26^a$ (5) | $-0.71^c$ (0) | **$0.86^c$ (0)** | $-0.70^c$ (0) | **$0.86^c$ (1)** | $0.82^c$ (3) | $0.34^b$ (0) |
| Cases | | | **$0.58^c$ (0)** | | $0.32^b$ (5) | $-0.35^b$ (2) | | |
| Deaths | $-0.77^c$ (1) | $0.58^c$ (0) | | $-0.74^c$ (0) | $0.73^c$ (0) | **$-0.85^c$ (3)** | $-0.74^c$ (2) | |
| Flights | **$0.94^c$ (1)** | | $-0.74^c$ (0) | | $-0.78^c$ (0) | $0.91^c$ (3) | $0.84^c$ (4) | $0.39^b$ (0) |
| Bicycles | $-0.85^c$ (3) | | $0.81^c$ (2) | **$-0.86^c$ (2)** | | $-0.85^c$ (5) | $-0.85^c$ (5) | $-0.42^c$ (0) |
| Train | $0.75^c$ (0) | $-0.32^b$ (0) | $-0.70^c$ (0) | $0.54^c$ (0) | $-0.56^c$ (0) | | **$0.84^c$ (0)** | |
| Stock | $0.73^c$ (0) | $0.34^b$ (5) | $-0.59^c$ (0) | $0.34^b$ (0) | $-0.46^c$ (0) | **$0.84^c$ (0)** | | $-0.33^b$ (4) |
| Vessel | **$0.47^c$ (2)** | $0.26^a$ (5) | $-0.33^b$ (3) | $0.45^c$ (3) | $-0.48^c$ (1) | $0.29^b$ (4) | $0.35^b$ (5) | |

direct negative correlation with traffic, flights, trains, and stocks and a lagged negative correlation with vessels. The lagged deaths have a positive correlation with bicycle searches.

Fig 11 highlights the correlation of traffic, flights, and stocks with the lagged corona deaths. The lag of the corona death is presented on the $x$-axis. In other words, it presents the correlation of the target variable with the corona deaths of $x$ weeks earlier. We can observe strong negative correlations of the target variables with the lagged corona deaths. All three variables show a similar trend where the negative correlation decreases as the lag of corona deaths increases. This indicates that, for example, the reduced traffic intensity due to an increase in corona death is almost recovered to the initial level after five weeks. In a similar way, flight

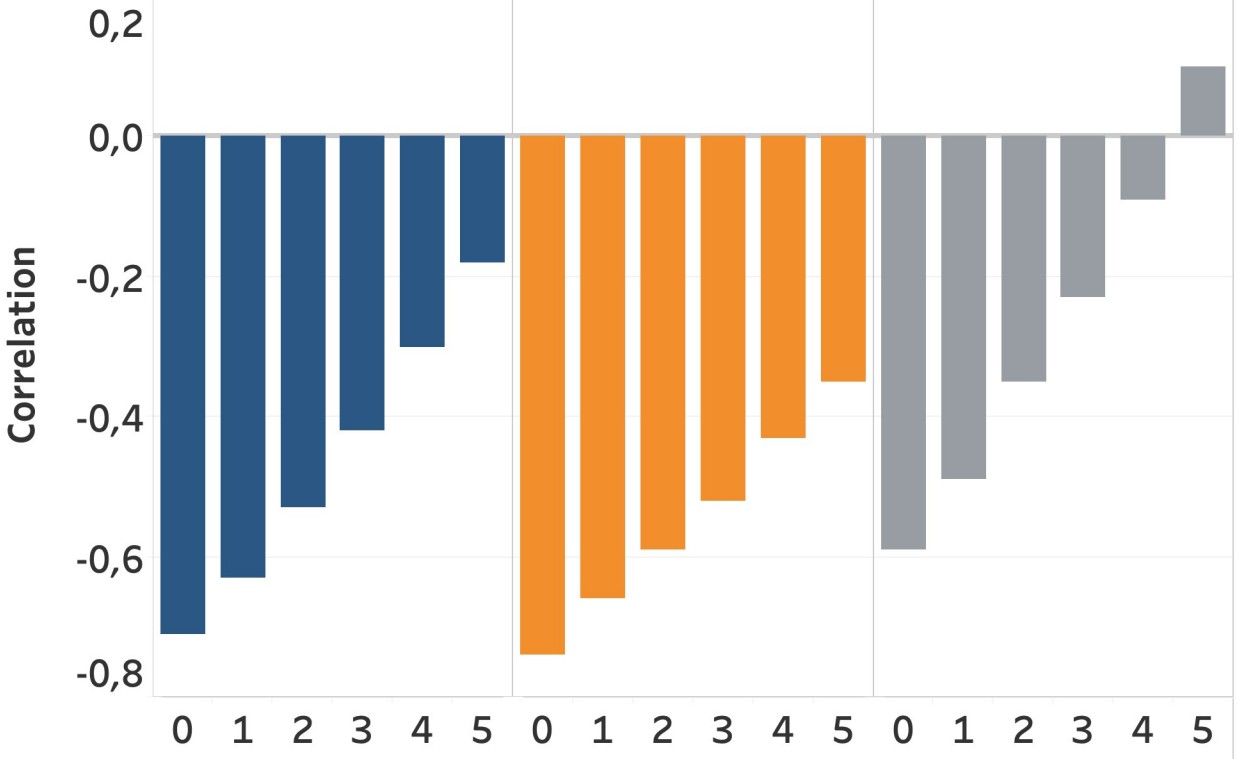

**Fig 11. Pearson correlation coefficient of traffic, flights, and stocks, with the lagged corona deaths variable on the $x$-axis.** It presents the correlation between the target variable and the corona deaths of $x$ weeks earlier. Traffic (blue), Flights (orange), and Stocks (grey).

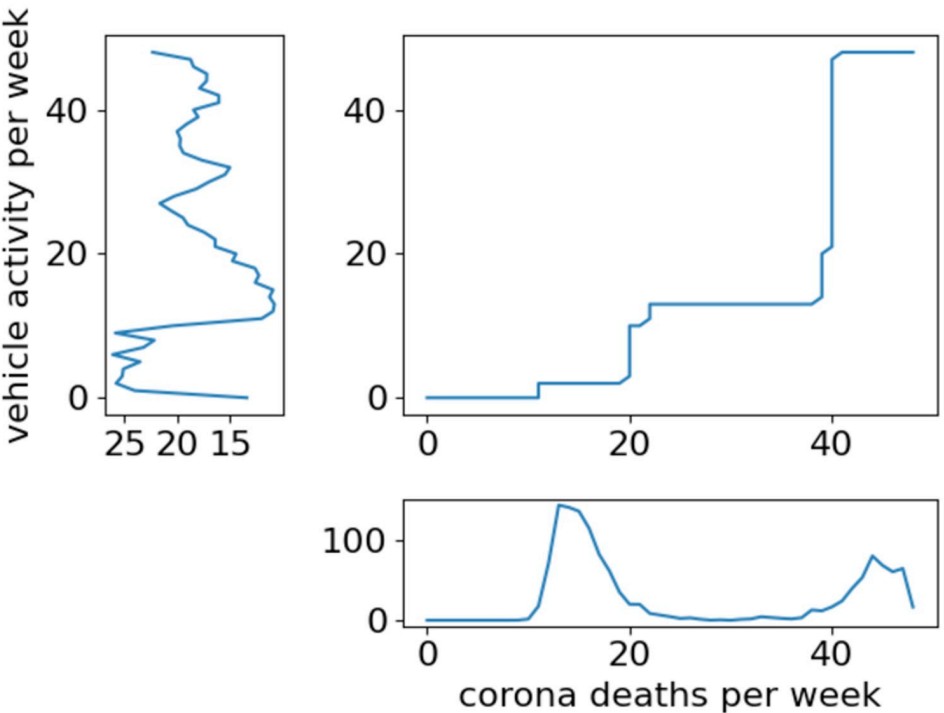

**Fig 12. Dynamic warping alignment plot between vehicle activity per week and corona deaths per week in the Netherlands.**

intensity recovers, but perceives a less strong recovery. This could potentially be due to the fact that flight are booked and scheduled in advance, resulting in an increased recovery time compared to road traffic. Moreover, flights potentially experience more corona restrictions as they often cross borders. Stock prices recover faster compared with traffic and flights.

Fig 12 shows the dynamic warping alignment plot of vehicle activity and corona deaths in the Netherlands. This indicates that the corona death increase is running ahead of the vehicle activity decline. It presents that the vehicle activity reacts slowly to the number of corona deaths.

### 4.3 Impact of Covid-19 measures

Fig 13 highlights the Corona deaths (top view) and the vessel, traffic, flight activity (bottom view), in relation to the largest changes in corona measures. A steep decline in both traffic and flights is visible after the first measures (closing all education, public areas, and sports) take effect. The decline in mobility continues as the maximum group sizes are further reduced to 3 people, and close contact professions are closed. Slightly after the first peek in deaths, a series of measures is reduced, resulting in a steady increase of mobility (3-5). A stable summer with a low number of death and a slow increase in traffic, follows after removing the maximum number of visitors for restaurants and reopening the gyms (6).

Surprisingly, the second wave of corona death with less strict measures that solely reduce the number of visitors and gradually maximizes the allowed group sizes from 6 to 2 (12-14), did not result in a decline in mobility. A rather stable pattern can be identified, deviating from the first wave.

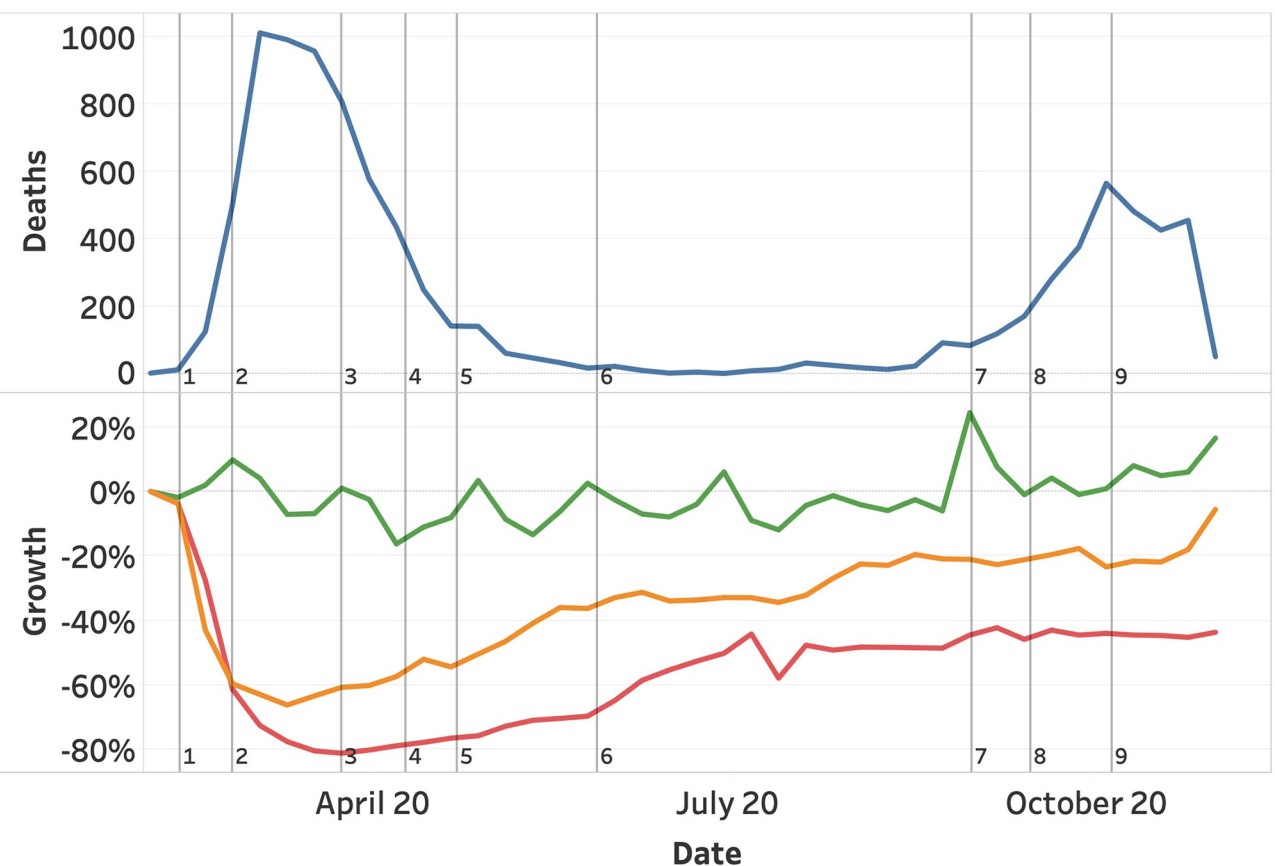

**Fig 13. Corona deaths with growth in vessel, traffic, and flights with respect to the major changes in corona measures.** Corona deaths (blue) and growth in vessel (green), traffic (orange), and flights (red). (1) closing all education, sport, and public areas. (2) reducing group sizes from 100 to 3 and closing close contact professions. (3) Opening sports under 18 years. (4) Opening close contact professions and sports for adults. (5) Open restaurants, increase group sizes from 3 to 6 and introduce mandatory face masks in public transport. (6) remove maximum visitors for restaurants and open gyms. (7) Reduce restaurant capacity to 30. (8) Close restaurants and max group sizes to 4. (9) Max group sizes to 2.

## 5 Discussion

In this paper, we presented and analyzed various relations between Covid-19, mobility, and stock-related data sources. Collecting and comparing such a wide range of data can only be done by making compromises. In this section, we will discuss our results and main design choices.

Overall, the Pearson correlations in Table 1 show significant relations between most variables. Especially Covid-19 deaths show strong correlations. This is in contrast with the Covid-19 cases, which show fewer significant and overall weaker correlations. This might be explained by deaths being registered more accurately or cases having a higher dependency on testing strategies.

A strong negative correlation between Covid-19 deaths and mobility with a low lag indicates that more deaths rapidly influences mobility in a negative way. The only exception is bicycle searches with has a strong positive correlation with a small lag of 2 weeks, indicating that bicycle searches do not rapidly spike after the corona deaths increase. Furthermore, we presented that traffic, flights and stock are negatively influenced by increased corona deaths. But that this correlation decreases linearly per week, where traffic is almost recovered to its original level within 5 weeks.

We present a strong correlation between death and the number of flights. However, we have to state that this possible causal relationship is difficult to measure due to the corona measures influencing this relation. Our reporting contains aggregations over multiple countries, being directly impacted by the measures taken in all countries.

Deaths seem to be a more reliable estimator compared with cases. This could potentially be related to different approaches for the registration of corona cases. However, the number of people in intensive care with corona might be a more reliable metrics compared with the number of cases and the number of death. Unfortunately, to the best of our knowledge, there is no publicly available dataset for the number of intensive care patients.

Regarding the flight data, we only have statistics available from February 2020 onwards. Ideally, we would cover the complete year, also as our other data sources did start in January 2020. However, our data sources do not allow us to go further back in time than six months. We could consider adding a ternary source for the flight data. However, this adds complexity and additional cost to the project.

Regarding the stock data, it has to be noted that stock markets do not operate in isolation. Markets in different time zones react to each other, resulting in some causal effects which are not identifiable in our data. We overcome this to a large extend by aggregation on a weekly level, but a small unmeasured causal effect remains.

In our current analysis, we do not take into account yearly seasonality. This seasonality would be relevant in making a more accurate comparison between the usage of the various modalities in 2020. For example, we know that bicycle usage correlates with the weather, showing an increase in summer usage each year. However, we did not include the seasonality aspect in our research as the majority of our data sources have a time window of less than a year. Including seasonal relationships would require a time window of at least two years to reliably estimate the impact. Therefore, a year from now, further research could focus on extending the dataset horizon to analyse seasonal patterns in the relationships.

In our current analysis, we do not include detailed statistics on a country or daily level. Ideally, we would have covered this analysis in more detail, but we decided to keep statistics compact and high-over. Exploring all variables on a lower level of detail would reduce the readability of this paper. Therefore, we would suggest further research based on the initial findings. Further research could focus on exploring the presented relation between death and flights on a country level, to better estimate the causal relationship. Furthermore, we presented strong relationships for Covid-19 deaths, in contrast with Covid-19 cases. Further research could could focus on the registration of Covid-19 cases and testing strategies, to evaluate how this weaker relationship is established.

## Supporting information

**S1 Text.**
(PDF)

## Author Contributions

**Conceptualization:** Robin Enrico van Ruitenbeek, Jesper Siem Slik, Sandjai Bhulai.

**Data curation:** Robin Enrico van Ruitenbeek, Jesper Siem Slik.

**Formal analysis:** Robin Enrico van Ruitenbeek, Jesper Siem Slik, Sandjai Bhulai.

**Funding acquisition:** Robin Enrico van Ruitenbeek, Jesper Siem Slik, Sandjai Bhulai.

**Investigation:** Robin Enrico van Ruitenbeek, Jesper Siem Slik, Sandjai Bhulai.

**Methodology:** Robin Enrico van Ruitenbeek, Jesper Siem Slik, Sandjai Bhulai.

**Project administration:** Robin Enrico van Ruitenbeek, Jesper Siem Slik, Sandjai Bhulai.

**Resources:** Robin Enrico van Ruitenbeek, Jesper Siem Slik, Sandjai Bhulai.

**Software:** Robin Enrico van Ruitenbeek, Jesper Siem Slik.

**Supervision:** Robin Enrico van Ruitenbeek, Jesper Siem Slik, Sandjai Bhulai.

**Validation:** Robin Enrico van Ruitenbeek, Jesper Siem Slik, Sandjai Bhulai.

**Visualization:** Robin Enrico van Ruitenbeek, Jesper Siem Slik.

**Writing – original draft:** Robin Enrico van Ruitenbeek, Jesper Siem Slik, Sandjai Bhulai.

**Writing – review & editing:** Robin Enrico van Ruitenbeek, Jesper Siem Slik, Sandjai Bhulai.

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
