## [Decision Letter · Decision Letter 0]

1 Sep 2021

PONE-D-21-22023

On the Relation between Covid-19, Mobility, and the Stock Market

PLOS ONE

Dear Dr. Slik,

Thank you for submitting your manuscript to PLOS ONE. After careful consideration, we feel that it has merit but does not fully meet PLOS ONE’s publication criteria as it currently stands. Therefore, we invite you to submit a revised version of the manuscript that addresses the points raised during the review process.

The paper requires further revisions towards prior literature, whereas the quantitative approach should be deeply extended.

We look forward to receiving your revised manuscript.

Kind regards,

Stefan Cristian Gherghina, PhD. Habil.

Academic Editor

PLOS ONE

Journal Requirements:

Reviewers' comments:

Reviewer's Responses to Questions

**Comments to the Author**

1. Is the manuscript technically sound, and do the data support the conclusions?

Reviewer #1: Yes

Reviewer #2: Yes

2. Has the statistical analysis been performed appropriately and rigorously? 

Reviewer #1: Yes

Reviewer #2: Yes

3. Have the authors made all data underlying the findings in their manuscript fully available?

Reviewer #1: No

Reviewer #2: No

4. Is the manuscript presented in an intelligible fashion and written in standard English?

Reviewer #1: Yes

Reviewer #2: Yes

5. Review Comments to the Author

Reviewer #1: 1-I strongly recommend adding a limitation and future research section and explain what was your limitations in research and suggest some offers for future research.

2-The authors must explain more details about the paper findings and clarify their conclusions for readers.

3-The literature needs to be strengthened with prior studies

Reviewer #2: 1. Overall, while the direct descriptive method and the quality of the presentation is convincing, from a descriptive point of view, authors do not show deep inferential statistical analysis based on their collected data.

2 In the introduction section, the authors stated, “The goal is to expose relations between the variables and understand them by using our data”. Which variables? Moreover, the authors may clarify the relations more in terms of statistical measures not just graphically.

3 In the discussion section, authors should justify why seasonality in the time series is ignored.

4 In the discussion section, the authors stated, “Our results show significant relations between most variables. Especially … 320

Here some statistical evidence is needed. Significance in terms of what?

6. PLOS authors have the option to publish the peer review history of their article (what does this mean?). If published, this will include your full peer review and any attached files.

Reviewer #1: No

Reviewer #2: No

---

## [Author Response · Author response to Decision Letter 0]

11 Nov 2021

We kindly refer to the attached file "Response to Reviewers.pdf".

---

## [Decision Letter · Decision Letter 1]

1 Dec 2021

On the Relation between Covid-19, Mobility, and the Stock Market

PONE-D-21-22023R1

Dear Dr. Slik,

We’re pleased to inform you that your manuscript has been judged scientifically suitable for publication and will be formally accepted for publication once it meets all outstanding technical requirements.

Kind regards,

Stefan Cristian Gherghina, PhD. Habil.

Academic Editor

PLOS ONE

Additional Editor Comments (optional):

Reviewers' comments:

Reviewer's Responses to Questions

**Comments to the Author**

1. If the authors have adequately addressed your comments raised in a previous round of review and you feel that this manuscript is now acceptable for publication, you may indicate that here to bypass the “Comments to the Author” section, enter your conflict of interest statement in the “Confidential to Editor” section, and submit your "Accept" recommendation.

Reviewer #2: All comments have been addressed

2. Is the manuscript technically sound, and do the data support the conclusions?

Reviewer #2: Yes

3. Has the statistical analysis been performed appropriately and rigorously? 

Reviewer #2: Yes

4. Have the authors made all data underlying the findings in their manuscript fully available?

Reviewer #2: Yes

5. Is the manuscript presented in an intelligible fashion and written in standard English?

Reviewer #2: Yes

6. Review Comments to the Author

Reviewer #2: Author(s) followed the reviewer's recommendations:

1) authors have shown some inferential statistical analysis based on their collected data as required

2) The introduction section has been modified

3) Discussion section has been clarified

7. PLOS authors have the option to publish the peer review history of their article (what does this mean?). If published, this will include your full peer review and any attached files.

Reviewer #2: No

---

## [Editor Report · Acceptance letter]

16 Dec 2021

PONE-D-21-22023R1 

On the Relation between Covid-19, Mobility, and the Stock Market 

Dear Dr. Slik:

I'm pleased to inform you that your manuscript has been deemed suitable for publication in PLOS ONE. Congratulations! Your manuscript is now with our production department. 

Kind regards, 

on behalf of

Dr. Stefan Cristian Gherghina 

Academic Editor

PLOS ONE